# Hyperspectral Inversion of Petroleum Hydrocarbon Contents in Soil Based on Continuum Removal and Wavelet Packet Decomposition

**Chaoqun Chen [1], Qigang Jiang [1,\*], Zhenchao Zhang [1], Pengfei Shi [1], Yan Xu [1], Bin Liu [1], Jing Xi [1] and ShouZhi Chang [2]** 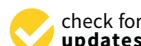

[1]   College of Geo-Exploration Science and Technology, Jilin University, Changchun 130026, China;
      chencq18@mails.jlu.edu.cn (C.C.); grayhelmet@foxmail.com (Z.Z.); shipf18@mails.jlu.edu.cn (P.S.);
      xuyanby@163.com (Y.X.); liubin17@mails.jlu.edu.cn (B.L.); xijing17@mails.jlu.edu.cn (J.X.)
[2]   School of Geomatics and Prospecting Engineering, Jilin Jianzhu University, Changchun 130118, China;
      changshouzhi@126.com
\*    Correspondence: jiangqigang@jlu.edu.cn

**Abstract:** Hyperspectral remote sensing is widely used to detect petroleum hydrocarbon pollution in soil monitoring. Different spectral pretreatment methods seriously affect the prediction and analysis of petroleum hydrocarbon contents (PHCs). This study adopted a combined spectral data preprocessing technique that improves the prediction accuracy of petroleum hydrocarbons in soil. We combined continuum removal and wavelet packet decomposition (CR–Daubechies 3 (db3)) to process the hyperspectral reflectance data of 26 soil samples in the oil production work area in China and judged the correlation between spectral reflectance and petroleum hydrocarbons in soil. Partial least squares regression was used to construct an optimal model for the inversion of PHCs in soil and the leave-one-out cross-validation was used to select the best factor number. The best model of soil petroleum hydrocarbon inversion was determined by comprehensively comparing the initial spectrum, db3 to high-frequency spectrum, db3 to low-frequency spectrum, after-continuum removal spectrum, CR-db3 to high-frequency spectrum, and CR-db3 to low-frequency spectrum comprehensively. The main contributions of this study are as follows: (1) three-layer decomposition with CR-db3 can improve the correlation between spectral reflectance and PHCs and effectively improve the sensitivity of the spectrum to PHCs; (2) the prediction accuracy of the high-frequency spectrum of wavelet packet decomposition for PHCs in soil is higher than that of low-frequency information; (3) the proposed petroleum hydrocarbon prediction model based on CR-db3 processed spectra to obtain high-frequency information is optimal (coefficient of determination = 0.977, root mean square error of calibration = 3.078, root mean square error of cross-validation = 4.727, root mean square error of prediction = 4.498, ratio of performance to deviation = 6.12).

**Keywords:** hyperspectral inversion; petroleum hydrocarbon content; CR-db3; wavelet packet decomposition

---

## 1. Introduction

Petroleum hydrocarbons are a complex mixture of hydrocarbons containing various hydrocarbons (n-alkanes, branched alkanes, cycloalkane, and aromatics) and a small amount of other organics [1]. With the increasing demand for oil and the gradual expansion of mining areas, oil leakage accidents have caused the risk of soil oil pollution [2]. Petroleum hydrocarbons will seriously affect the soil environmental quality, change the physical and chemical characteristics of the soil by reducing soil permeability, and endanger the development of vegetation roots in soil [3,4]. Therefore, rapidly

detecting and predicting the content of petroleum hydrocarbons in soil are of great importance to reduce losses and harm.

Professionals have mainly used field sampling to monitor traditional petroleum hydrocarbon pollution in soil and measure total petroleum hydrocarbons in the laboratory, which is time-consuming and difficult, lacks universal applicability, and cannot identify and monitor the petroleum hydrocarbon contents (PHCs) in soil on a large scale [5]. The rapid development of hyperspectral remote sensing provides a fast and inexpensive method for petroleum hydrocarbon pollution analysis [6–9]. In the late 1980s, Cloutis used visible–near-infrared spectroscopy to study the reflection spectrum characteristics of hydrocarbons and proved that a certain relationship is found between petroleum and spectrum [10]. However, the spectral absorption characteristics of petroleum hydrocarbon are weakened because of the influence of moisture, temperature [11], soils at various particle sizes [12], and other factors. Scholars have used spectral data processing methods, including continuum removal, standard normal energy transform, differential derivation, and principal component analysis to extract soil petroleum hydrocarbon information, amplify the spectral absorption characteristics of petroleum hydrocarbon, and accurately predict the PHCs [13–16]. Ren [12] chose the 360–600 nm band spectrum for pretreatment and used the first derivative as spectral pretreatment in building models for predicting soil petroleum hydrocarbon concentration on the basis of visible–near-infrared spectroscopy and spectral analysis (coefficient of determination ($R^2$) = 0.65, root mean square error (RMSE) of prediction = 60.58 g/kg). Scafutto [13] used principal component analysis and partial least squares regression (PLSR) to examine the near and shortwave infrared spectral data of several mineral substrates impregnated with crude oils (°APIs 19.2, 27.5, and 43.2), diesel, gasoline, and ethanol ($R^2$ > 0.9). Chakraborty [14] used both first derivative of reflectance and discrete wavelet transformations to preprocess the oil soil spectra. Three clustering analyses and three multivariate regression methods were used for pattern recognition and to develop the petroleum predictive models, and stepwise multiple linear regression based on discrete wavelet transformations had the best prediction effect ($R^2$ = 0.94, RPD = 3.97). Most scholars have used linear models to predict the spectrum of soil petroleum hydrocarbon pollution, where PLSR has a better prediction effect [17–19].

The absorption characteristics of petroleum hydrocarbons in the soil spectrum are relatively weak, and the information related to petroleum hydrocarbons is mostly concentrated in high-frequency data [20]. This study used the soil samples in the oil production area as the research object to efficiently decompose the high-frequency and low-frequency data of the soil spectral information and maximize the preservation of the spectral characteristics of petroleum hydrocarbons in soil. The hyperspectral data were preprocessed by combining continuum removal and wavelet packet decomposition (CR–Daubechies 3 (db3)). We combined leave-one-out cross-validation to establish PLSR, and selected the optimal estimation model of PHCs by comparing the accuracy of the spectral prediction models of different methods to provide a feasible analysis for the subsequent inversion of large-scale soil petroleum hydrocarbon pollution.

## 2. Materials and Methods

### 2.1. Determination of PHCs

The research area was located in Zhaoyuan County, Daqing City, Heilongjiang Province. It is an intensive oil production operation area in Daqing Oilfield [21]. The soil texture is mainly clay and sandy soil [22]. Soil organic matter contents are higher than in other places in China, and the contents mostly range from 2% to 4% [23]. The soil collection area was located in the northwest of Zhaoyuan County, and the well site soil with oil production wells was selected along the oil production road. We used the base of the oil production well as the center, set the radius to 50 cm, and collected 26 surface soil samples (0–15 cm) in this area (Figure 1). Soil samples were placed in the laboratory and were naturally dried. After removing the debris, such as plant roots and gravel, samples were passed through a 1 mm sieve. PHCs (C10–C14) in soil were determined using a gas chromatograph GC7900

in accordance with the US-EPA method SW-846-8015B. Table 1 shows the statistical characteristics of PHCs from 26 soil samples, and the detailed information of the samples are in Appendix A.

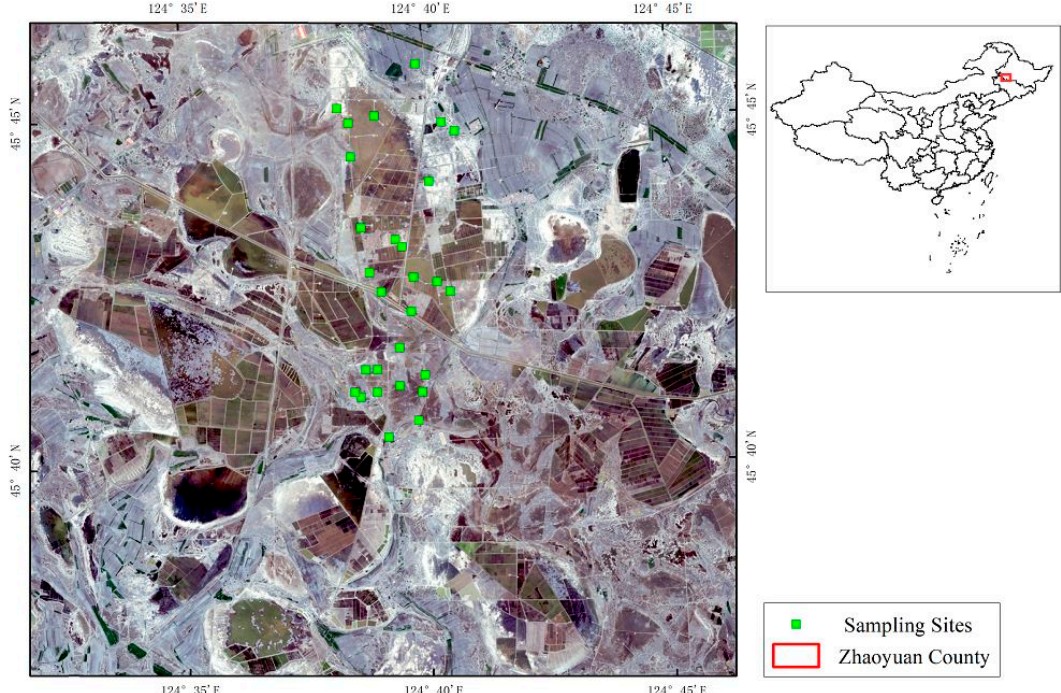

**Figure 1.** Distribution of sampling points.

**Table 1.** Statistical characteristics of PHCs from 26 soil samples.

|  | Sample Number | Max | Min | Average | Standard Deviation |
|---|---|---|---|---|---|
| PHCs (g/kg) | 26 | 85 | 0.008 | 15.082 | 20.569 |

### 2.2. Spectral Measurement

The spectral reflectance of the soil samples was determined using an Analytical Spectral Device (ASD) FieldSpec Pro in the laboratory (Figures 2 and 3). The obtained information ranged from 350 nm to 2500 nm, the light source was a built-in halogen lamp, the field angle of the probe was 25°, and the sampling intervals were 1.4 (350–1000 nm) and 2 nm (1000–2500 nm). The surface of the soil sample was approximately 10 cm from the probe. Each sample was scanned for 15 times, the average value was calculated, and the larger noise bands at 350–399 and 2451–2500 nm were removed as the initial reflection spectrum of the soil samples (Figure 4).

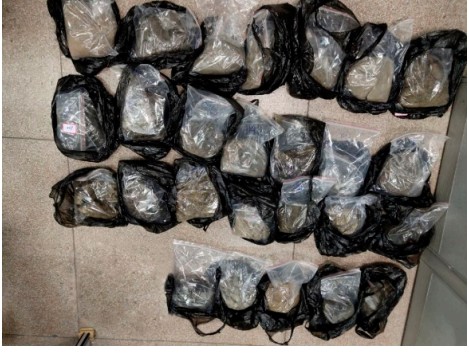

**Figure 2.** Soil samples.

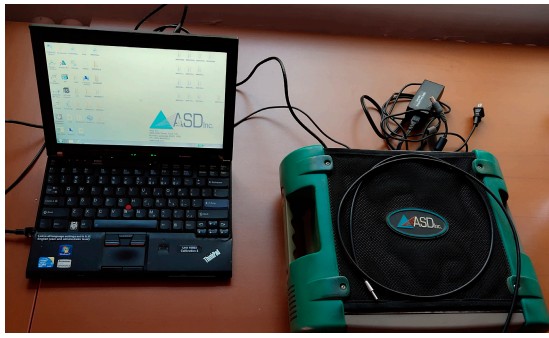

**Figure 3.** FieldSpec Pro ASD.

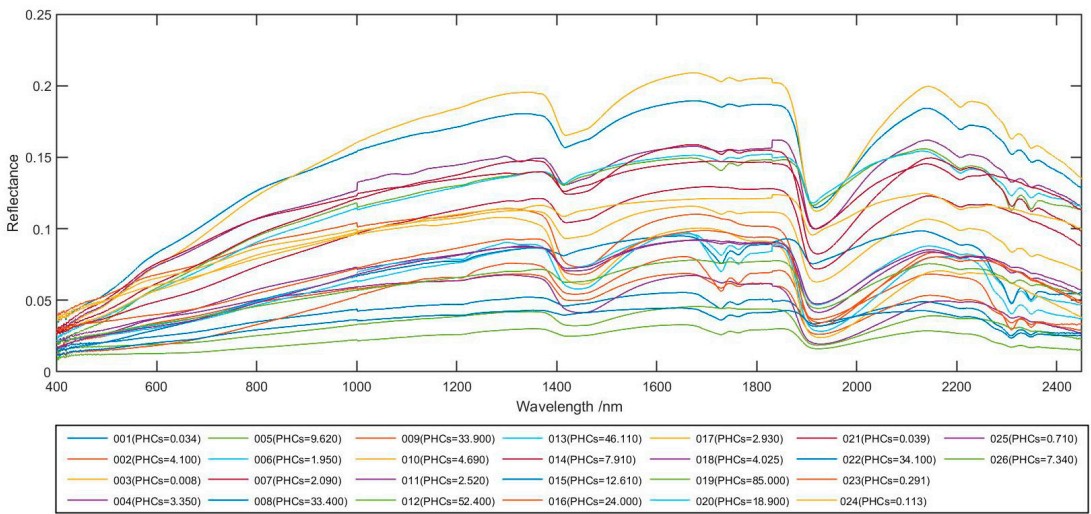

**Figure 4.** Initial reflection spectrum of the soil samples.

## 2.3. Method

### 2.3.1. Continuum Removal

The soil spectrum is a combination of various information, and continuum removal is introduced to suppress the background information and highlight the absorption characteristics of the target. After continuum removal, the spectral reflectance is normalized to 0–1. The principle is described as follows: A point straight line connects the peak point, and an envelope is a polyline with the outer angle of the peak point greater than 180°. Initial spectral reflectance divides the relative reflectance curve on the envelope to obtain continuum removal reflectance [24,25]. The spectrum after continuum removal can substantially amplify the spectral absorption bands and enhance the contrast between the spectral bands [26].

### 2.3.2. Daubechies 3 (db3) Wavelet Packet Decomposition

Wavelet packet analysis is a detailed signal analysis method based on orthogonal wavelet decomposition and can characterize the local information in the time and frequency domains [27]. Wavelet packet analysis is more refined for signal extraction, and the resolution of the high-frequency part of its decomposition is better than wavelet analysis [28]. The smooth error introduced by the third-order db3 wavelet is difficult to be observed, and beneficial information can be completely extracted during signal decomposition.

Taking the three-layer wavelet packet decomposition as an example, the high-frequency and low-frequency parts are obtained after the first wavelet packet decomposition. The low-frequency and

high-frequency parts are semi-decomposed simultaneously in the second decomposition, as shown in Figure 5.

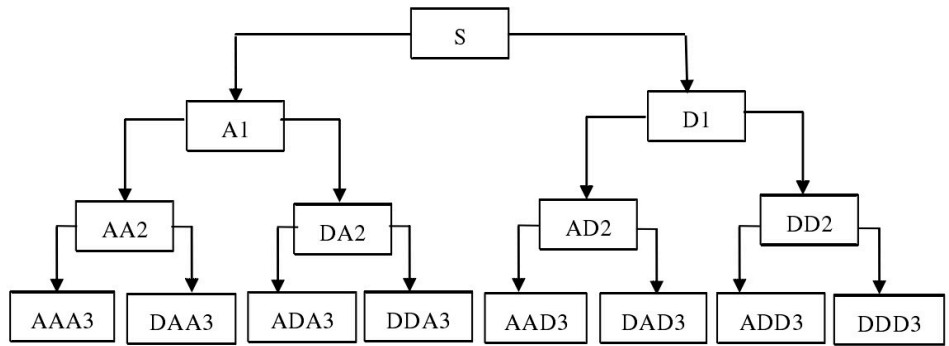

**Figure 5.** Wavelet packet decomposition tree.

In this figure, *A* is the low frequency, *D* is the high frequency, and the number at the end indicates the number of layers of wavelet decomposition. Original data *S* can be expressed as Equation (1):

$$S = AAA3 + DDA3 + ADA3 + DDA3 + AAD3 + ADA3 + ADD3 + DDD3 \tag{1}$$

In this study, the initial and after-continuum removal spectra of the soil samples were subjected to db3 wavelet decomposition to obtain low-frequency and high-frequency components. The data for one soil sample was used as an example:

Initial and after-continuum removal spectra were used as references (Figure 6a–d) to compare and analyze the spectral characteristics of different pretreatment methods. These methods included db3 to low-frequency (Figure 6b) and db3 to high-frequency spectra (Figure 6c) obtained by decomposing the initial spectrum through db3 wavelet packet decomposition and CR-db3 to low-frequency spectrum (Figure 6e) and CR-db3 to high-frequency spectrum (Figure 6f) obtained by combining continuum removal and wavelet packet decomposition.

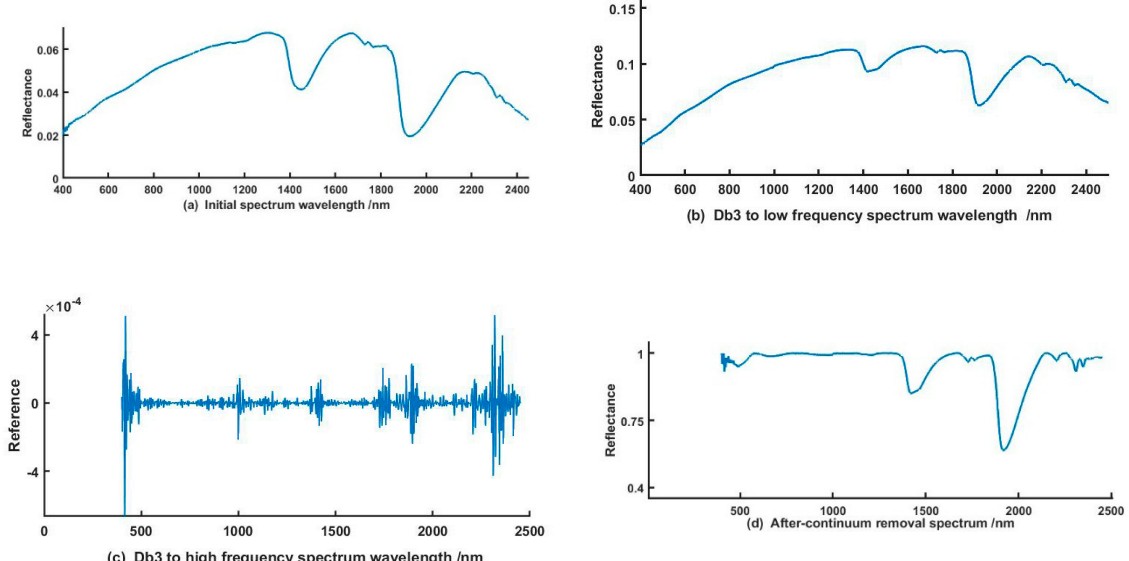

**Figure 6.** *Cont.*

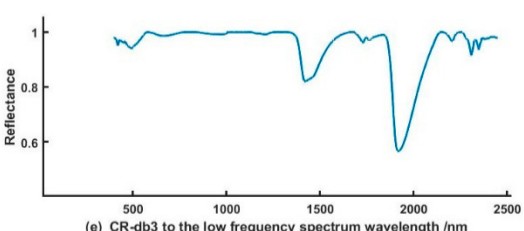 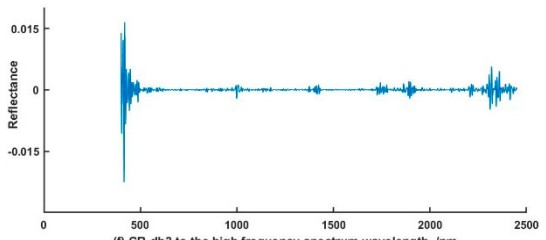

**Figure 6.** Db3 wavelet packet decomposition.

### 2.3.3. Pearson Correlation Analysis

This study aimed to explore the relationship between different spectral data preprocessing techniques and PHCs, and study the effect of pretreatment methods on the spectral sensitivity of petroleum hydrocarbons through the correlation analysis of initial spectrum, db3 to high-frequency spectrum, db3 to low-frequency spectrum, after-continuum removal spectrum, CR-db3 to the high-frequency spectrum, and continuum CR-db3 to the low-frequency spectrum and the PHCs in soil. Pearson correlation was used to judge the linear correlation between spectral reflectance and PHCs (Equation (2)).

$$r = \frac{\sigma_{xy}^2}{\sigma_x \sigma_y} = \frac{\sum [(X_i - X)(Y_i - Y)]}{\sqrt{\sum (X_i - X)^2} \sqrt{\sum (Y_i - Y)^2}} \tag{2}$$

where $r$ is the correlation coefficient, $X$ is the reflectance of the corresponding band, $Y$ is the PHCs, and $\sigma_{xy}^2$ is the covariance of $X$ and $Y$. In this study, the bands with high correlation ($r > |0.6|$) between the spectral reflectance of petroleum hydrocarbons were selected for PLSR modeling to predict the PHCs.

### 2.3.4. PLSR

PLSR analysis is suitable for regression modeling under the condition that the number of samples is less than the number of variables [29]. The number of bands used to establish PLSR is significantly more than the sample data and is widely used in the inversion study on petroleum hydrocarbons in soil using visible–near-infrared spectroscopy. PLSR combines the characteristics of principal component analysis, canonical correlation analysis, and linear regression analysis methods during modeling. However, PLSR is relatively different from other regression analyses. The model considers the correlation between independent variables and the relationship between independent and dependent variables when extracting principal components [30,31]. PLSR modeling was performed on the "The Unscrambler X 10.4" to predict PHCs [32].

In this study, 19 soil sample data were selected as the test set, and the remaining data were used as the validation set. PLSR was established by taking the reflectance of each band of the spectrum (400–2450 nm) as the independent variable and petroleum hydrocarbon content as the dependent variable, and was evaluated on the basis of $R^2$, root mean square error of calibration (RMSEC), root mean square error of cross-validation (RMSECV), root mean square error of prediction (RMSEP), and ratio of performance to deviation (RPD) [33]. The closer $R^2$ is to 1 and the smaller the RMSE, the more stable the model [5,6]. RPD < 1.4 indicates that the model has poor predictive power, 1.4 ≤ RPD ≤ 2 indicates that the model can roughly estimate the samples, and RPD ≥ 2 indicates that the model has strong predictive power [34,35]. The formulas are expressed as follows:

$$R^2 = 1 - \frac{\sum\limits_{i=1}^{n} (y_{ib} - y_{ia})^2}{\sum\limits_{i=1}^{n} (y_{ib} - y_i)^2} \tag{3}$$

$$RMSE = \sqrt{\frac{\sum\limits_{i=1}^{n}(y_{ib} - y_{ia})^2}{n}} \qquad (4)$$

$$RPD = \frac{\sigma}{\sqrt{\sum\limits_{i=1}^{n}(y_{ia} - y_{ib})^2 / n}} \qquad (5)$$

where n is the number of samples used during the prediction, $y_{ia}$ is the prediction value, $y_{ib}$ is the reference value for sample i, and $\sigma$ is the standard deviation of the measured reference values.

## 3. Results and Discussion

### 3.1. Spectral Analysis of Petroleum Hydrocarbons in Soil

By observing initial reflection spectra of the soil samples with different PHCs (Figure 4), we found that the morphological characteristics of these spectral curves were basically the same. In the whole wavelengths (400–2450 nm), the reflectance of petroleum hydrocarbons was not high, and the reflectance significantly decreased with the increase in PHCs in soil. The absorption bands were apparent after continuum removal to process the initial spectrum (Figure 7). According to the research of Cécile, the absorption valleys at 1400–1430, 1918–1925, and 2205 nm are related to OH⁻ contained in clay minerals [25]. Absorption valleys were found near 400–550, 1200–1220, 1700–1730, 1750–1770, 2309–2311, and 2346–2350 nm. At the same time, the absorption valleys moved down with the increase in PHCs in soil. This finding indicates that these absorption valley bands contained response bands for petroleum hydrocarbons, and spectral reflectance can reveal the PHCs to a certain extent. This is consistent with the research findings of Chakraborty [14], Rosa [16], Wang [36], and other scholars, and the sensitive bands of petroleum hydrocarbons they found were also in these ranges. Using db3 wavelet to perform three-layer decomposition on the initial spectrum and after-continuum removal spectrum, the low-frequency (Figure 6b,e) and undecomposed spectra (Figure 6a,d) were similar. This finding is because low-frequency data belong to the image frame and account for most of the information. The high-frequency information obtained through wavelet packet decomposition was rich, and the absorption bands were evident (Figure 6c,f) because they reflected the detailed information of the image.

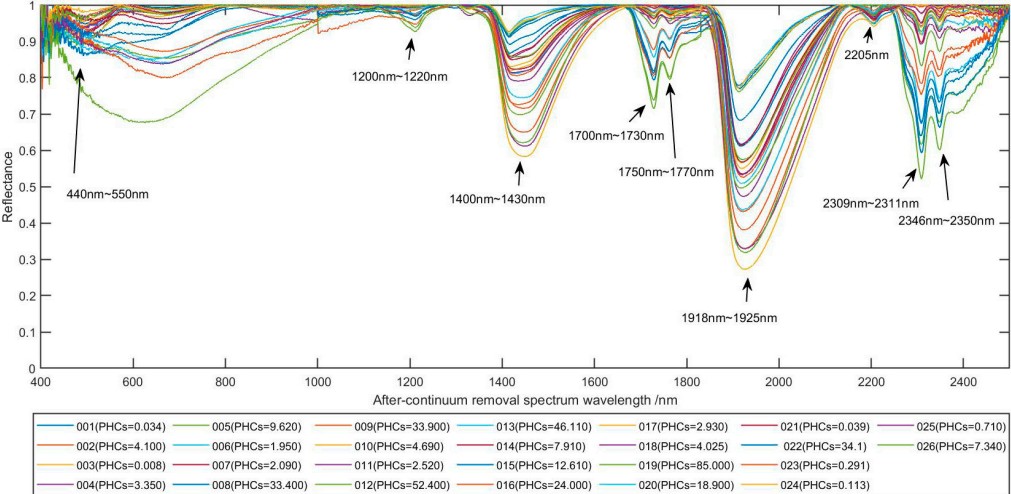

**Figure 7.** Continuum removal spectrum.

### 3.2. Correlation Analysis of Soil Spectrum and PHCs

This study calculated the Pearson correlation coefficient of PHCs and the reflectance of each band (400–2450 nm) in terms of the initial spectrum, db3 to high-frequency spectrum, db3 to low-frequency spectrum, after-continuum removal spectrum, CR-db3 to high-frequency spectrum, and CR-db3 to low-frequency spectrum to obtain the sensitive bands of PHCs in soil. As shown in Figure 8a,b, the initial and db3 to low-frequency spectra were inversely related to PHCs. As shown in Figure 4, the spectral reflectance decreased with the increase in PHCs. After continuum removal, the spectral noise was largely eliminated, and the correlation between reflectance and PHCs increased (Figure 8d–f). The sensitivity of db3 wavelet packet decomposition high-frequency spectral information to PHCs was significantly higher than that of low-frequency information, where CR-db3 to high-frequency spectrum was the highest (r = −0.809–0.808) (Figure 8f) by comparing Figure 8b,c,e, and f.

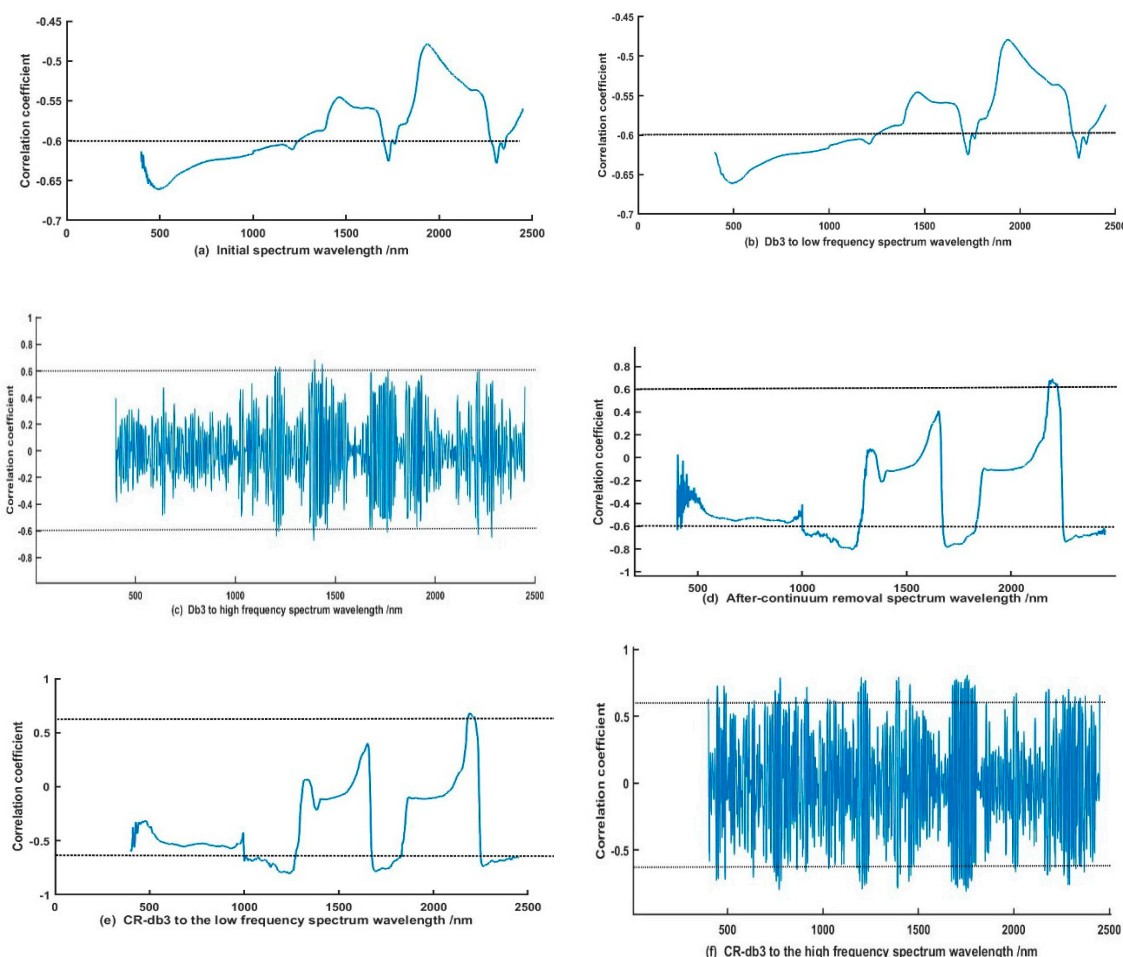

**Figure 8.** Correlation coefficient between PHCs and soil spectrum.

At the same time, referring to Table 2, the set of important variables obtained during the PLSR modeling process basically corresponded to the bands with high correlation (r > | 0.6 |). It can be seen that these bands with high correlation play an important role in explaining PHCs.

### 3.3. Petroleum Hydrocarbons Spectrum Prediction Model

The study used different preprocessing techniques to process the spectral data to establish PLSR for estimating PHCs. As shown in Table 2, the resulting model had the lowest accuracy, with $R^2$ of 0.422, and RMSEC, RMSECV, RMSEP were the largest using the unprocessed spectrum for modeling. The low-frequency data obtained through db3 wavelet three-layer decomposition of the initial spectrum

contained little petroleum hydrocarbon information. Thus, the estimation model performed by this spectrum preprocessing method had small predictive power (RPD = 1.06). The RMSE values of the spectral model after continuum removal were more than 10g/kg and the models had the ability to predict the content of PHCs (RPD > 3). The prediction modeling accuracy of PHCs after continuum removal was better than that of the initial spectrum. The $R^2$ values of db3 wavelet packet decomposition to high-frequency spectrum were greater than 0.9, and the RMSE was low. Among the spectra, the CR-db3 to high-frequency spectral prediction model had the best prediction method, highest accuracy, and strongest prediction ability ($R^2$ = 0.977, RMSEC = 3.078, RMSECV = 4.727, RMSEP = 4.498, RPD = 6.16).

The high-frequency spectrum obtained through the three-layer decomposition of db3 wavelet packet can improve the correlation between spectral reflectance and PHCs (Figure 8c,f) and significantly improve the diagnostic accuracy of the model. Db3 wavelet packet decomposition can strip weak information because of the low PHCs in soil, which belong to detailed information. At the same time, continuum removal can highlight the characteristics of target information. So CR-db3 can make the predicted value close to the actual value to a great extent. Figure 9 shows the comparison between the measured and predicted values of PHCs using CR-db3 to high-frequency spectrum. The modeling and verification sample points were roughly distributed around the straight line y = x.

**Table 2.** Modeling accuracy of different spectral prediction models.

| Spectral Preprocessing Models | Important Variables | $R^2$ | RMSEC (g/kg) | RMSECV (g/kg) | RMSEP (g/kg) | RPD |
|---|---|---|---|---|---|---|
| Initial spectrum | 456–524, 1886–189, 1995–2033, 2278–2371 nm | 0.422 | 12.358 | 16.565 | 22.528 | 1.01 |
| Db3 to low-frequency spectrum | 502–524, 1887–189, 1997–203, 2285–2368 nm | 0.423 | 11.360 | 15.813. | 20.534 | 1.06 |
| Db3 to high-frequency spectrum | 483, 697, 701, 872–883, 1387–1440, 1552–1565, 1898–1932, 2202–2220 nm etc. | 0.818 | 8.800 | 9.189 | 12.804 | 2.33 |
| After-continuum removal spectrum | 1186–1254, 1682–1843, 2202–2226, 2250–2448 nm | 0.918 | 5.888 | 7.971 | 10.662 | 3.29 |
| Db3 to low-frequency spectrum | 469–474, 835–869, 1185–1307, 2224–2232 nm | 0.951 | 4.578 | 5.147 | 6.952 | 3.38 |
| Db3 to high-frequency spectrum | 441–443, 447–483, 686–740, 1423–1441, 1683–1716, 2257–2296, 2400 nm etc. | 0.977 | 3.078 | 4.727 | 4.498 | 6.16 |

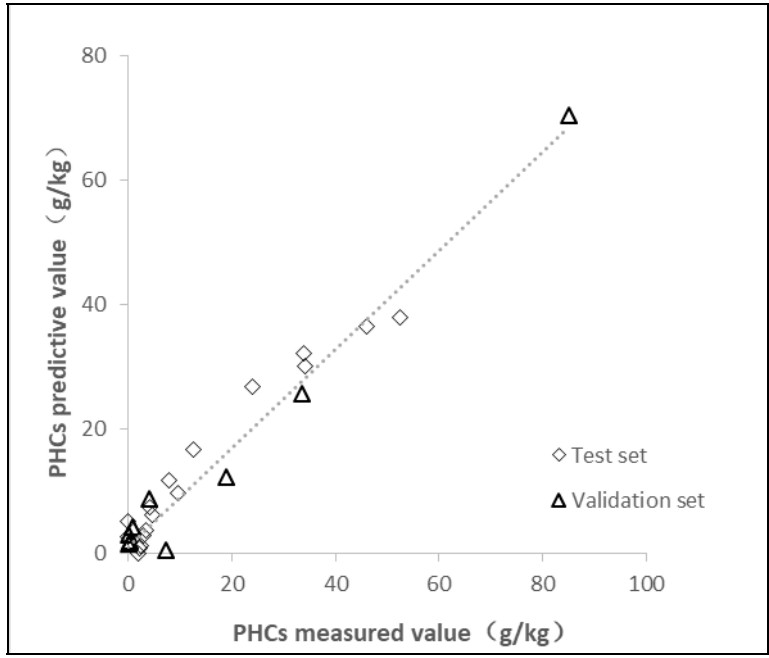

**Figure 9.** Scatter plot of PHC prediction model based on CR-db3 to high-frequency spectrum.

## 4. Conclusion

Petroleum hydrocarbons easily leak during extraction, transportation, and refinement and damage the sustainable development of the environment when they enter the soil. Therefore, monitoring soil petroleum hydrocarbon pollution in real time with the help of hyperspectral inversion is extremely important [37]. The soil petroleum hydrocarbon spectrum in Zhaoyuan County performed mathematical transformation through continuum removal and wavelet packet decomposition (CR-db3). The correlation coefficients of the spectral reflectance of six pretreatment operations and PHCs were calculated, and the corresponding PLSR models were established to predict the petroleum hydrocarbon content. The results are summarized as follows:

(1) CR-db3 three-layer decomposition can improve the correlation between the spectral reflectance and PHCs and effectively improve the spectrum sensitivity to PHCs.

(2) Wavelet packet decomposition can improve the accuracy of the PHC prediction model in soil, where the accuracy of the obtained high-frequency spectrum modeling is higher than that of the low-frequency data.

(3) Obtaining high-frequency information based on the CR-db3 processed spectrum can improve the prediction accuracy of PHCs. The PHC model constructed by this preprocessing method is optimal ($R^2$ = 0.977, RMSEC = 3.078, RMSECV = 4.727, RMSEP = 4.498, RPD = 6.16).

**Author Contributions:** C.C. developed the original idea for the study and completed the manuscript. Q.J. and Z.Z. revised the paper. P.S., Y.X., and B.L. contributed to data collection. J.X. and S.C. helped complete program code. All authors have read and agreed to the published version of the manuscript.

**Funding:** This research was funded by China Geological Survey, "Remote sensing geological survey and monitoring of the state of mine development in China" (202012000000180606).

**Acknowledgments:** Thanks are given to the Editor and the reviewers for their helpful comments and criticisms, which have greatly improved the presentation and accuracy of the paper.

**Conflicts of Interest:** The authors declare no conflict of interest.

## Appendix A

**Table A1.** 26 soil samples of PHCs.

| Number | Longitude Latitude | | PHCs (mg/kg) |
|---|---|---|---|
| 001 | 124°39′52.17″E | 45°45′45.95″N | 34.3 |
| 002 | 124°39′0.50″ E | 45°45′1.25″ N | 4100 |
| 003 | 124°38′14.48″ E | 45°45′8.30″ N | 8.08 |
| 004 | 124°38′27.90″ E | 45°44′55.42″ N | 3350 |
| 005 | 124°38′30.29″ E | 45°44′26.33″ N | 9620 |
| 006 | 124°38′41.50″ E | 45°43′24.91″ N | 1950 |
| 007 | 124°38′50.88″ E | 45°42′46.08″ N | 2090 |
| 008 | 124°39′4.99″ E | 45°42′29.06″ N | 33400 |
| 009 | 124°39′23.42″ E | 45°43′14.21″ N | 33900 |
| 010 | 124°39′31.58″ E | 45°43′8.08″ N | 4690 |
| 011 | 124°39′45.21″ E | 45°42′41.35″ N | 2520 |
| 012 | 124°39′41.43″ E | 45°42′12.00″ N | 52400 |
| 013 | 124°40′23.31″ E | 45°44′55.42″ N | 46110 |
| 014 | 124°40′39.19″ E | 45°44′47.29″ N | 7910 |
| 015 | 124°40′6.45″ E | 45°44′4.12″ N | 12610 |
| 016 | 124°40′14.13″ E | 45°42′37.05″ N | 24000 |
| 017 | 124°40′30.04″ E | 45°42′28.58″ N | 2930 |
| 018 | 124°39′53.38″ E | 45°41′2.20″ N | 4025 |
| 019 | 124°39′26.28″ E | 45°41′40.69″ N | 85000 |
| 020 | 124°38′58.33″ E | 45°41′22.04″ N | 18900 |
| 021 | 124°38′44.03″ E | 45°41′22.00″ N | 39.3 |
| 022 | 124°38′57.63″ E | 45°41′2.60″ N | 34100 |
| 023 | 124°39′25.78″ E | 45°41′7.70″ N | 291 |
| 024 | 124°38′37.23″ E | 45°40′58.38″ N | 113 |
| 025 | 124°39′10.87″ E | 45°40′23.44″ N | 710 |
| 026 | 124°39′48.02″ E | 45°40′37.36″ N | 7340 |

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
