# Peer review of "Hyperspectral Inversion of Petroleum Hydrocarbon Contents in Soil Based on Continuum Removal and Wavelet Packet Decomposition"

_sustainability, doi:10.3390/su12104218_

Round 1
Reviewer 1 Report
The manuscript introduced a method that can improve the correlation between the spectral reflectance and petroleum hydrocarbons contents. The authors concluded that continuum removal and wavelet decomposition to high frequency could achieve this goal. Overall, the topic is great, and the conclusions are well supported by the method and analysis.
As a revised version, the manuscript provides accurate information and has much better readability. However, some low-level mistakes, such as “leave one method” cross-validation, indicate that the authors might need to improve their understanding of statistics. I suggest a major revision before accepted for publication. I suggest the authors work as a group and double-check every single statistic concept that appeared in the manuscript to make sure there is no misrepresentation.
Specific comments/suggestions:
Figure 3 please adjust the height/width ratio of the figure. The picture was obviously distorted. This was not taken care of in the last revision.
Line 73: Leave one out, not leave one method
Line 227: Leave one out, not leave one method
Line 229: how could R-square be 4.81?
Author Response
Dear Reviewer:
Please see the attachment, and detailed replies or explanations on your comments are given in the attachment.
Thank you very much!
Best regards!
Yours sincerely,
Chaoqun Chen

Reviewer 2 Report
Manuscript was improved. However, additional correction is necessary.
My comments are below.
Lines 12-13 actually, you did not propose preprocessing technique, you applied already developed methods to your data.
Line 18 - leave one out method
Lines 22-28: I suggest switching (1) and (2)
Line 47 - how are petroleum hydrocarbon pollution and soil organic matter related?
line 99 - do you mean these noisy regions were removed?
Lines 113-114 - it does not clear how you reached this conclusion.
line 166 - typical correlation analysis??what is that? do you mean parametric?
lines 165-167 the phrasing is wrong
Does Figure 7 presents the data for one sample or it is average result for all samples?
Lines 189-190 do you have any interpretation for these absorption features?
As Figure 4 is analyzed in Results section it would be better to place it there.
Lines 190-192 this statement is not supported neither by the text nor by the figure 4.
Line 205 - incorrect reference to Figure 1 which shows geographical location of test area.
Line 209 - something wrong with numbers
Lines 215-216 What does it mean?
Lines 218-219 PLSR does not require prior removement of spectral bands based on their correlation with modeled parameter.
Lines 223-228 describe methods not results.
Author Response

(The authors gave the same response as above.)

Round 2
Reviewer 2 Report
The manuscript has been improved. All my comments have been successfully addressed.
I just have several additional comments.
line 189 - do you mean part of the spectrum?
line 235 - DPR is RPD?
line 261 - PLSR models?
Author Response
Dear Reviewer:
Please see the attachment, and detailed replies or explanations on your comments are given in the attachment.
Thank you very much!
Best regards!
Yours sincerely,
Chaoqun Chen

This manuscript is a resubmission of an earlier submission. The following is a list of the peer review reports and author responses from that submission.
Round 1
Reviewer 1 Report
The paper presents a description of an attempt to use the db3 wavelet decomposition method to build PLRS prediction models of PHC content in soil with the use of fibre optic, field NIR spectrometer.
The topic of the work is very interesting and with a potential.
The research purpose is original and well-formulated.
The state of the art, methodology, results and their discussion are presented well but with shortcomings and major objections:
In the literature review, several works from classics of soil PHC determination by NIR are missing. The description of the methodology lacks a description of the physicochemical properties of soils used in the research. The description of the soil type is of little relevance to the work. It would not have to be a full soil analysis, but at least texture, organic matter content is necessary. The reader does not know if the soil samples were scanned dry (how dry?), moist? The reader of the work does not know anything about how the samples were sampled, let alone how they were treated and how they were prepared for scanning. There's only information about the angle and distance The authors do not specify which device was used in gas chromatography. There is no information about the size of the test and validation set in the PLSR. The authors used coefficients such as r2 and RPD to describe and analyse the quality of the obtained models. Shouldn't there be R2 instead of r2? A commonly used quality parameter in chemometrics is RMSE. The authors did not use it. The results and discussions did not make any reference to the obtained RPD coefficients. Are they high or low? Are the models useful for these values and what are the limitations? The authors themselves point out, what is valuable, the shortcomings in the work, which aresmall number of soil samples and PHCs large difference. The reader does not receive convincing arguments to ignore these deficiencies.
Other comments:
Figures 7 and 8 - Require graphic cleanup and improved readability. Line 201: The abbreviation DPR is incomprehensible Very often at work there is a lack of space at units e.g. at nm. - The example is line 74. Instead of 399nm it should be 399 nm. A mysterious graphic sign appears in Table 3, it was probably supposed to be a comma.

Reviewer 2 Report
Article presents analysis of spectral reflectance of soils polluted by Petroleum Hydrocarbons.
The novelty of the work is that it uses more advanced spectral data preprocessing technique which allows increasing the accuracy of models obtained with preprocessed data.
My comments can be found below.
Check your English.
Why do you use word "inversion" in title and lines 59, 94, etc. What do you mean by it?
The introduction should be rewritten.
What is the main aim of your research?
To find out indicative bands for petroleum hydrocarbons? Why are you not satisfied with those obtained in previous studies?
To test more advanced method of spectral preprocessing and to acquire model with higher accuracy? In this case the title of the article needs to be changed. And you should also add the information on the accruracy of the models acquired in previous studies.
Also it is not clear why did you choose the method of db3 wavelet decomposition? What are the advantages of this method compared to others. Add more references on that.
Methods section-
please add the scheme of preprocessing combinations
Results section should start with proper analysis of obtained spectral data
Why did not you include spectral reflectance of non-polluted soils in the analysis?
Sepcific comments:
lines 7-8 Sentence needs to be rephrased
line 17-18 – Which method do these bands correspond to?
Line 19 – spectral reflectance cannot be developed
Table 1 – It may be better to move full table to supplementary material and leave only descriptive statistics (mean, max, min, sd , etc). Anyway, you need to comment on table, not just put it in the text.
Figure 4 – reflectance spectrum
Table 2 – This table is not necessary as these are the basics. If you want to leave it, at least add the reference.
Line 108 – add reference on previous studies you mentioned
Line 106-112 describe the result not the methodology. In Materials and Methods you should provide the description of the methods.
Figure 7 – a,b,c,d etc should be described in the caption to the figure. All the graphs should be alined.
Line 141, 143 – Which image did you refer to? You worked with spectral curves. May be you meant samples?
Lines 139-143 should be in Results section.
Line 144 - why do you need to calculate correlation before plsr modelling? plsr does not require prior choice of indicative wavelengths. it does it in the process.
Line 156 – What is calibration coefficient? Do you mean R2 which is determination coefficient. You need to use ajusted R2 to correct for the large number of predictors.
Lines 164 – 167 – It is better to describe all the combinations properly in Materials and Methods section and do not to list them all here in one sentence. And this information should be in Materials and Methods not in the Results.
Line 176 – test probability should be explained in Materials and Methods
Table 3 - You need to analyse the table not just mention it. Do all the methods have common characteristic wavelengths? What does ~ mean in the table?
Line 188-190 - do you mean that in plsr model you also included characteristic absorbtion bands with highest correlation from different preprocessing combinations?
Table 4 - presented data is for the model before or after cross-validation?
include more information on models (complexity). were some other wavelengths included in models?
Line 221-223 - What do you mean by the expantion of effective band? Did you do it? How?
Line 239 - why are these patrticular wavelengths mentioned in conclusion?
Reviewer 3 Report
The manuscript introduced a method that can improve the correlation between the spectral reflectance and petroleum hydrocarbons contents. The authors concluded that continuum removal and wavelet decomposition to high frequency could achieve this goal. Overall, the topic is great, and the conclusions are well supported by the method and analysis. While this work provides a reference for the subsequent hyperspectral remote sensing monitoring of petroleum hydrocarbons pollution in 24 soil, I think at this stage the manuscript is pretty much just a foundation and very preliminary step of mapping for petroleum hydrocarbons pollution. I recommend the authors to keep working on the project and then submit a manuscript with more comprehensive results. Therefore, I do not recommend publication at this phase.
Specific comments/suggestions:
Line 46 to 50 Please re-organize the logic to improve readability
Table 3 style issue: 400nm ~ 1244nm 、 1702nm~ 1745nm
Replace “~” with “ –”, and “、” with “,” because “~” usually means “about”.
Line 156 “ RPD and r2 were the smallest, indicating that the model prediction accuracy was improved”. Please check if this information is correct (largest?).
Figure 3 please adjust the height/width ratio of the figure.
Figure 8 please increase the size of the words to make sure they are readable.
Line 248 move period to line 247